# Metabolite Profiling and Transcriptome Analysis Explain the Difference in Accumulation of Bioactive Constituents in Taxilli Herba from Two Hosts

**DOI:** 10.3390/genes14051040

**Published:** 2023-05-04

**Authors:** Jiahuan Yuan, Nan Wu, Zhichen Cai, Cuihua Chen, Yongyi Zhou, Haijie Chen, Jia Xue, Xunhong Liu, Wenxin Wang, Jianming Cheng, Li Li

**Affiliations:** 1College of Pharmacy, Nanjing University of Chinese Medicine, Nanjing 210023, China; yuanjiahuan1027@163.com (J.Y.);; 2Jiangsu Province Engineering Research Center of Classical Prescription, Nanjing 210023, China; 3College of Pharmacy, Guangxi University of Chinese Medicine, Nanning 530220, China

**Keywords:** Taxilli Herba, hosts, transcriptomic, metabolomic, biosynthesis network

## Abstract

Taxilli Herba (TH) is a semi-parasitic herb and the host is a key factor affecting its quality. Flavonoids are the main bioactive constituents in TH. However, studies on the difference in accumulation of flavonoids in TH from different hosts are vacant. In this study, integrated transcriptomic and metabolomic analyses were performed on TH from *Morus alba* L. (SS) and *Liquidambar formosana* Hance (FXS) to investigate the relationship between the regulation of gene expression and the accumulation of bioactive constituents. The results showed that a total of 3319 differentially expressed genes (DEGs) were screened in transcriptomic analysis, including 1726 up-regulated genes and 1547 down-regulated genes. In addition, 81 compounds were identified using ultra-fast performance liquid chromatography coupled with triple quadrupole-time of flight ion trap tandem mass spectrometry (UFLC-Triple TOF-MS/MS) analysis, and the relative contents of flavonol aglycones and glycosides were higher in TH from SS group than those from the FXS group. A putative biosynthesis network of flavonoids was created, combined with structural genes, and the expression patterns of genes were mostly consistent with the variation of bioactive constituents. It was noteworthy that the UDP-glycosyltransferase genes might participate in downstream flavonoid glycosides synthesis. The findings of this work will provide a new way to understand the quality formation of TH from the aspects of metabolite changes and molecular mechanism.

## 1. Introduction

Taxilli Herba (TH), a member of the Loranthaceae family, is derived from dried stems and branches with leaves of *Taxillus chinensis* (DC.) Danser [1]. TH is a well-known traditional Chinese herb that has been used for the treatment of symptoms such as soreness and weakness of the waist and knees [2,3]. According to phytochemical research, chemical components including flavonoids [4], organic acids [5], and volatile oils [6], etc., have been identified. As a semi-parasitic plant, it has hundreds of host species [7]. However, there is no clear requirement for hosts of TH in the Chinese pharmacopoeia [1]. At present, the majority of medical products being utilized and circulated on the market come from multiple hosts, with uneven quality [8]. The two most frequent hosts among them are *Morus alba* L. (SS) and *Liquidambar formosana* Hance (FXS). Currently, the study on TH from different host sources is mainly divided into three section, namely, the content determination of major flavonoids [9,10,11,12,13], pharmacological effects [14,15,16], and the distinction of TH from different hosts using various approaches [17,18]. The host is a crucial element impacting the quality of TH, according to the studies accessible so far [10,11,15]. 

As the main bioactive constituents, the contents of flavonoids in TH from different hosts are different, despite the fact that studies of flavonoids’ biosynthesis pathways and regulatory networks in model plants have been fully deciphered [19,20]. In the study of flavonoids in *Arabidopsis thaliana*, the genes involved in flavonoid skeleton biosynthesis (*PAL*, *CHI*, *4CL*, *F3H*, *HCT*, *FLS*, etc.) have been completely identified, and several flavonoid-modified genes have been isolated [21,22]. The characterization of MYB transcription factors of other transcription factors such as bHLH and NAC, as well as flavonols, anthocyanins and proanthocyanidins, has also been undertaken [23]. At the same time, flavonoids play a variety of roles in existing plants, including affecting cell wall synthesis, pollen development, resistance to low temperature, and resistance to ultraviolet [24,25,26,27]. Studies on the accumulation of quercetin and kaempferitrin in *Arabidopsis thaliana* showed that even the same substance can have different regulatory mechanisms under different environmental conditions [28]. However, to the best of our knowledge, the underlying molecular mechanisms by which host variations may lead to differences in the synthesis and accumulation of bioactive constituents remain unclear due to the paucity of transcriptomic information. Only by fully understanding the formation of bioactive constituents in TH can we provide a rational basis for the host selection of TH. 

RNA-sequencing (RNA-Seq) has the advantage of high throughput, wide coverage, and precise sequencing of cDNA [29], and it is essential for the mining of species resilience genes [30], biosynthesis, and the identification of regulatory major genes [31]. On the one hand, changes in metabolites can directly affect phenotypic changes by reflecting the true physiological conditions of the organism. However, the genetic process affecting the phenotype cannot be fully explained by metabolomic detection alone. Conversely, transcriptome expression does not necessarily show phenotypic change. Therefore, the combination of transcriptomic and metabolomic analyses can be used to explore biological problems from the two levels of “cause” and “effect”, deeply analyze the macro development process of biological systems, and explain the complexity and integrity of biological processes [32,33,34]. 

In this work, we investigated the molecular mechanisms involved in the biosynthesis of the chemical components of TH from SS and FXS, using an integrated transcriptomic analysis and metabolomic approach. Bioinformatics analysis was used to select out and analyze differentially expressed genes (DEGs). A putative biosynthesis network was created, and genes implicated in flavonoid biosynthesis were chosen. This work would not only facilitate the elucidation of flavonoid biosynthesis in TH from different hosts, but also provide available information for the selection of hosts.

## 2. Materials and Methods 

### 2.1. Plant Materials

Two parallel TH samples from *Morus alba* L. (No.2022011901-2022011903) and *Liquidambar formosana* Hance (No.2022011904-2022011906) were collected from Wuzhou of Guangxi Province in China, each with three biological replicates. The botanical origins of the materials were authenticated by Professor Xunhong Liu (Nanjing University of Chinese Medicine). Voucher specimens were deposited in the laboratory of Chinese Medicine Identification, Nanjing University of Chinese Medicine. Stems and branches with leaves were collected and washed with phosphate-buffered saline, and frozen in liquid nitrogen for metabolomic analysis and transcriptomic analysis.

### 2.2. Metabolite Profiling Analysis of Bioactive Constituents in Taxilli Herba

Three biological replicates of samples in two groups were powdered and sieved through a 50 mesh. Approximately 0.5 g of powder was extracted by sonication with 15 mL 50% methanol for 30 min at room temperature. After centrifugation at 13,000 r/min for 10 min, the supernatant was passed through a 0.22 μm membrane (Jinteng laboratory equipment Co., Ltd., Tianjin, China). The solutions were stored at 4 °C before ultra-fast liquid chromatography coupled with triple quadrupole-time of flight tandem mass spectrometry (UFLC-Triple TOF-MS/MS) analysis [35].

All samples were analyzed using a UFLC system (Shimadzu, Kyoto, Japan) with the separation conducted by an Agilent ZORBAX SB-C_18_ column (4.6 mm × 250 mm, 5 μm) at 30 °C. The mobile phase composed of methanol: acetonitrile (1:1, *v*/*v*) (A) and 0.4% (*v*/*v*) formic acid aqueous with a flow rate at 1.0 mL/min was set as follows: 0–5 min, 2–6% A; 5–6 min, 6–10% A; 6–8 min, 10–15% A; 8–12 min,15–18% A; 12–18 min, 18–21% A; 18–21 min, 21–23% A; 21–26 min, 23–25% A; 26–30 min, 25–27% A; 30–33 min, 27–40% A; 33–38 min, 40–50% A; 38–40 min, 50–2% A; 40–45 min, 2–2% A. The injection volume was 10 μL [35]. 

The Triple TOF^TM^ 5600-MS/MS system (AB SCIEX, Framingham, MA, USA) equipped with an ESI source was used to obtain MS data in negative mode. The optimized MS parameters were as follows: the scanning range of TOF MS/MS: *m*/*z* 50–1500; the ion source temperature (TEM): 600 °C; the curtain gas (CUR): 40 psi; the nebulization gas (GS1): 60 psi; the auxiliary gas (GS2): 60 psi; the collision energy: –10 V; the ion spray voltage floating (ISVF): 4500 V; the declustering potential voltage: –100 V [35].

The UFLC-Triple TOF-MS/MS data were analyzed by PeakView 1.2 software. The precise molecular mass was determined with MS^1^ and the fragment information was obtained with MS^2^. For the identification of compounds, one method was to compare with a previously established chemical composition database, and verify by the retention time and the mass spectrometry data of the standards. The second method was to speculate by combining the database and the relevant literature. At the same time, the relative peak area of metabolites was extracted, and the average value of three samples was taken as the relative content of one metabolite.

### 2.3. Transcriptome Analysis of Taxilli Herba

#### 2.3.1. RNA Preparation

Total RNA was extracted from TH samples using Plant RNA Purification Reagent (Invitrogen, Carlsbad, CA, USA) following the manufacturer’s instructions, and genomic DNA was eliminated using DNase I (TaKara Bio. Inc., Changping, Beijing, China). Then, the integrity and purity of the total RNA were determined with a 2100 Bioanalyser (Agilent Technologies, Inc., Santa Clara, CA, USA) and quantified using the Nanodrop 2000 (NanoDrop Thermo Scientific, Wilmington, DE, USA). Only high-quality RNA samples (OD260/280 = 1.8–2.2, OD260/230 ≥ 2.0, RIN ≥ 8.0, 28S: 18S ≥ 1.0, > 1 μg) were used to construct a sequencing library.

#### 2.3.2. Library Preparation and Sequencing 

RNA purification, reverse transcription, and library construction and sequencing were conducted by Shanghai Bio-pharm Biotechnology Co., Ltd. (Shanghai, China) following the manufacturer’s instructions. Firstly, an Illumina TruSeqTM RNA sample preparation Kit (Illumina, SD, CA, USA) was used to prepare the transcriptome library using 1 μg of total RNA. Briefly, mRNA was isolated and enriched from total RNA by A-T base pairing with ployA using magnetic beads with Oligo (dT). Following random fragmentation with fragmentation buffer, tiny fragments of about 300 bp were subsequently separated by magnetic bead screening. Subsequently, under the action of reverse transcriptase, a strand of cDNA was synthesized in reverse by adding random primers. Thereafter, two-strand synthesis was performed to create stable double-stranded cDNA. End Repair Mix was introduced to this base and utilized to link the adaptor by complementing the double-stranded cDNA into flat ends. Finally, library electrophoresis was performed on 2% Low Range Ultra Agarose with 200~300 bp cDNA as target fragments, which was amplified for 15 cycles using Phusion DNA polymerase (NEB, Ipswich, MA, USA). The paired-end RNA-seq sequencing library was sequenced using an Illumina NovaSeq 6000 sequencer (2 × 150 bp read length) after being quantified by TBS380. 

#### 2.3.3. De Novo Assembly and Function Annotation

First, the bases quality, error rate, and distribution were examined using fastX_toolkit (http://hannonlab.cshl.edu/fastx_toolkit/, accessed on 3 February 2022). Raw data were filtered using Sickle (https://github.com/najoshi/sickle, accessed on 3 February 2022) and SeqPrep (https://github.com/najoshi/sickle, accessed on 3 February 2022) to remove low-quality sequences and contaminated adaptors. High-quality clean data were acquired with fastp (https://github.com/OpenGene/fast, accessed on 19 April 2022) followed by de novo assembly with Trinity (https://github.com/trinityrnaseq/trinityrnaseq, accessed on 19 April 2022). NCBI non-redundant protein sequences (NR); the Swiss PROT protein sequence database (Swiss-Prot); homologous protein family (Pfam); Clusters of Orthologous Groups of proteins (COG); Gene Ontology (GO); and the Kyoto Encyclopedia of Genes and Genomes (KEGG) were used to acquire function annotations of assembled transcripts using BLAST+ (ftp://ftp.ncbi.nlm.nih.gov/blast/executables/blast+/2.9.0/, accessed on 13 May 2022). BLAST2GO (version 2.5.0) was used to annotate GO functions, while KOBSA (version 2.1.1) was used to find linked metabolic pathways.

#### 2.3.4. Identification and Analysis of Differentially Expressed Genes

The expression level of each transcript was calculated according to the fragments per kilobases per million reads (FPKM), and the abundance was quantified using RSEM software. Furthermore, genes with |log_2_(foldchange)| ≥ 1, *p*-value < 0.05, and false discovery rate (FDR) < 0.01 were considered as DEGs. In addition, to determine the biological functions and pathways involved, DEGs underwent the enrichment analysis of GO and KEGG. Meanwhile, metabolomics combined with genes related to flavonoid biosynthesis were further analyzed. 

#### 2.3.5. qRT-PCR Verification

qRT-PCR was applied to quantify the relative mRNA expression patterns of the 9 genes (*CCoAOMT*, *CYP98A*, *bglB*, *CAD*, *CSE*, *HCT*, *FLS*, *4CL*, *F3H*) which were involved in secondary metabolites synthesis. Total RNA was extracted from samples using the Trizol total RNA isolation kit, according to the manufacturer’s instructions. The primers were designed using Primer 3.0 (https://bioinfo.ut.ee/primer3-0.4.0/, accessed on 3 June 2022). cDNA pools from the total RNA were synthesized for the qRT-PCR using HiScript Q RT SuperMix for qPCR (Vayme Biotech Co., Ltd., Nanjing, China). The 20 μL reaction volume included 10 μL 2 × ChamQ SYBR Color qPCR Master Mix, 0.8 μL of each primer (μM), 0.4 μL 50 × ROX Reference Dye 1, and 2 μL cDNA. The PCR procedure was set as follows: 95 °C for 5 min, followed by 40 cycles at 95 °C for 5 s, primer annealing at 55 °C for 30 s, and extension at 72 °C for 40 s. Actin was used as the reference standard, and the relative gene expression level was calculated using the qRT-PCR system (ABI 7500 qRT-PCR system) using the $2^{-\Delta \Delta \mathrm{Ct}}$ method. Primers for the PCR are shown in Appendix A. 

#### 2.3.6. Data Processing

The graphs were created by Origin (version 2022). All experimental data were analyzed by *t*-test (SPSS, Beijing, China, version 21.0), and a *p*-value < 0.05 indicated significant difference. The results were represented by the mean values and standard deviations of three biological replicates.

## 3. Results

### 3.1. The Change in Metabolic Profiling and Metabolite Content in Taxilli Herba 

The metabolic profiles of TH from SS and FXS were investigated according to the previously established UFLC-Triple TOF-MS/MS method [35]. The variations in their metabolic profiles and the alterations in the relative levels of the metabolites were the subject of further study. The representative base peak chromatograms of two groups were shown in Appendix A. A total of 81 compounds were identified based on retention times, precise molecular weights, and MS^2^ characteristic fragments, combined with comparisons with standards, database searches, and reports in the relevant literature (Appendix A). The relative peak area of the identified metabolites was extracted, and the average of the relative peak areas of three independent samples in each group was calculated as the relative content of a metabolite (Appendix A). Five different metabolites, including monotropein, quercitrin, procyanidin B2, procyanidin B1, and procyanidin B2 3″-O-gallate, were screened out in TH from two hosts with VIP > 1 and *p* < 0.05 as the criteria.

Figure 1A,B illustrated some fluctuation in the relative levels of metabolites in two TH groups. Since flavonoids are the main active constituents in TH, we investigated the total accumulation of flavonoid aglycones and flavonol glycosides in two groups. The results demonstrated that the total accumulation of both varied. Flavan-3-ol and flavonol glycosides were more prevalent among them and accumulated more in TH from SS (Figure 1C,D). Similar patterns were seen in the accumulation of other constituents (Figure 1E). In detail, dihydroflavone, dihydroflavanol, flavonol, flavone, flavane, isoflavone, and their corresponding glycosides were obtained and analyzed in this experiment. As the main active constituent, the relative content of quercitrin in TH from SS was significantly higher than that from FXS (Figure 1F). The heatmap of the metabolite profile (Appendix A) revealed that samples from SS contained slightly more of the majority of aglycones and glycosides than those from FXS.

### 3.2. De Novo Transcriptome Assembly and Sequence Analysis 

In the present study, the original sequence of TH from SS ranged from 51,805,610 to 57,061,100, which was optimized to a high-quality sequence ranging from 50,022,130 to 55,502,034, and the mapped ratio was 78.15%; the original sequence of TH from FXS ranged from 45,975,326 to 50,792,870, which was optimized to a high-quality sequence ranging from 44,058,022 to 48,557,952, and the mapped ratio was 78.08%. Each transcriptome sample had up to 6.32 GB of clean data. The Q30 (sequences with a sequencing error rate of less than 0.1%) base percentage was 93.37%, the Q20 base percentage was 97.55% and the average content of GC was 48.82% (Appendix A). Overall, the RNA-seq data had high quality, which could be used for further analysis. The de novo assembly of all clean data was performed using Trinity (version v2.8.5) and the assembly results were optimized. With a total of 57,203 unigenes and 103,690 transcripts retrieved, the average N50 lengths were 2035 bp and 2195 bp, respectively. In general, the sequencing results were sufficient to be used for subsequent analysis. 

### 3.3. Functional Annotation and Classification

Following assembly, the unigenes were annotated in six databases (GO, KEGG, COG, NR, Swiss-Prot, and Pfam), yielding a total of 21,197, 10,215, 21,644, 25,791, 17,828, and 17,499 sequences, respectively (Appendix A). 

The GO terms were distributed into 52 groups (Appendix A), which were further classified into three categories, namely, biological process, cellular component, and molecular function. The top GO terms were cellular process (9584) and metabolic process (8554) in the biological process category. In the cellular component category, cell part (10,391) and membrane part (7528) were major GO terms. In the molecular function category, binding (11,864) and catalytic activity (10,319) ranked at the top of the terms. As shown in the KEGG annotation (Appendix A), 10215 (17.86%) genes were assigned to five main categories after being significantly matched in the KEGG pathway database. The largest category was metabolism (2739), followed by genetic information processing (1937) and environmental information processing (375). Among them, 146 genes that were involved in the biosynthesis of other secondary metabolites were classified and annotated. Appendix A listed the secondary metabolites pathways, with the major pathways being phenylpropanoid biosynthesis (79), flavonoid biosynthesis (17), flavone and flavonol biosynthesis (3), and isoflavonoid biosynthesis (1).

A total of 3319 DEGs were discovered, including 1726 up-regulated genes and 1593 down-regulated genes in TH from FXS (Figure 2). After being mapped to the GO database, enrichment analysis was performed on all DEGs. Aspartyl-tRNA aminoacylation, aspartate-tRNA ligase activity, and the response to chitin were the three most significant functions, and the function with the highest number was transcription regulator activity (Figure 3A). Among the top 20 items of KEGG enrichment analysis (Figure 3B), most DEGs were enriched in pathways, including ribosome (79), plant–pathogen interaction (29), and sesquiterpenoid and triterpenoid biosynthesis (7). Among them, nine DEGs were enriched in phenylpropanoid biosynthesis. 

### 3.4. Analysis of Genes and Metabolites Involved in Flavonoid Biosynthesis in Taxilli Herba 

A total of 93 genes involved in flavonoid biosynthesis were screened out (Appendix A). Here, the genes were mapped to the specific positions of active constituents, and the relative content of metabolites and the expression of genes in the synthetic pathway were compared and analyzed jointly (Figure 4) [36]. 

In this study, two phenylalanine ammonia-lyases (*PALs*) were annotated. The FC values ranged close to 1 despite the fact that the expression of one was up-regulated and the other was down-regulated in TH from SS. 4-coumarate-CoA ligase (*4CL*) participated in the biosynthesis of flavonoids in the downstream steps of the pathway, which was a substrate for the synthesis of various flavonoids constituents. Nine putative *4CL*s were identified. Six of these showed an up-regulated trend, while the others showed a down-regulated tendency, which might help to explain why the metabolic group results for TH from SS were slightly greater than those from FXS in terms of the relative content of phenylpropanoids and organic acids (Figure 1E). Caffeic acid 3-O-methyltransferase (*COMT*) and cinnamoyl-CoA reductase (*CCR*) were DEGs. Dihydroflavonoids can generate dihydroflavonol compounds under the action of flavanone-3-hydroxylase (*F3H*). The expression of F3H was up-regulated in TH from SS, which was consistent with the fact that the relative contents of dihydroflavone and dihydroflavonol in TH from SS were higher than those in TH from FXS. One of the identified flavonol synthases (*FLSs*) had a |Log_2_ FC| of 3.77, although it was incongruent with the slight variation in the relative content of quercetin in TH from two hosts. Similarly, anthocyanidin synthase (*ANS*), anthocyanidin reductase (*ANR*), dihydroflavonol 4-reductase (*DFR*) and leucoanthocyanidin reductase (*LAR*) involved in the catechin synthesis pathway were all down-regulated in TH from SS, inconsistent with the high relative levels of metabolites in this pathway. The changes in UDP-glycosyltransferase expression were shown in Appendix A. Twenty-two UDP-glycosyltransferases were up-regulated and twenty-eight were down-regulated in TH from SS, but none of the differences in expression were significant, which was a reasonable explanation for the small differences in the overall content of flavonoid glycosides. 

Spearman correlation analysis was used to analysis the differential components and differential genes. As shown in Appendix A, quercitrin was significantly positively correlated to *XPOT;E1.11.1.7* (TRINITY_DN29103_c0_g1) and *E1.11.1.7* (TRINITY_DN31463_c0_g1), and significantly negatively correlated to *E1.11.1.7* (TRINITY_DN3430_c0_g2), *K22395* (TRINITY_DN4783_c0_g1), *CYP73A* (TRINITY_DN9699_c0_g1), and *PRDX6* (TRINITY_DN15784_c0_g1). Monotropein showed the opposite significant relationship with *E1.11.1.7* (TRINITY_DN3430_c0_g2) and *E1.11.1.7* (TRINITY_DN31463_c0_g1), respectively. Procyanidin B2 3″-O-gallatefructose had a significantly positive correlation with *CCR* (TRINITY_DN20850_c0_g2). Five metabolites had a weak correlation with *COMT* (TRINITY_DN14175_c0_g2). The aforementioned findings indicated that the biosynthesis of quercitrin may be related to the regulation of peroxidase to some extent, although no pertinent studies have been reported on this, making it worth in-depth investigation in future studies as well.

### 3.5. Confirmation of the Expression of Related Genes Using qRT-PCR

Nine genes involved in the flavonoid biosynthesis pathway were selected for qRT-PCR. As shown in Figure 5, the expression trends of *HCT* (TRINITY_DN3882_c0_g1), *FLS* (TRINITY_DN1730_c0_g1), and *4CL* (TRINITY_DN29174_c0_g1) were consistent with the transcriptome data, while *CCoAOMT* (TRINITY_DN6080_c0_g1), *CYP98A* (TRINITY_DN5837_c0_g1), *bglB* (TRINITY_DN9022_c0_g1), *CAD* (TRINITY_DN3782_c0_g1), *CSE* (TRINITY_DN192_c0_g1), and *F3H* (TRINITY_DN3827_c0_g1) showed an opposite trend. The possible reason for the above phenomenon was that the fold change of expression of these genes between the two groups was less than 1. Among them, the key structural genes FLS and HCT were up-regulated in TH from SS. 

### 3.6. Transcription Factors 

The class of proteins called transcription factors (TFs) are widely found in living organisms and have an activating or blocking effect on gene expression through specific functional domains that recognize and bind to cis-acting elements in the upstream regulatory regions of genes. A DNA-binding domain, a trans-activating domain, and other signal sensing domains make up the majority of TFs. Different transcription factor families can be formed from TFs based on their functional domains. A total of 113 TFs were predicted for DEGs using TF prediction and analysis (Appendix A), with the AP2/ERF family and NAC family (both 19.47%) having the highest percentage, followed by the WRKY family (14.16%), MYB superfamily (9.73%), and bHLH family (5.31%). The management of the plant secondary metabolism, especially the synthesis of major medicinal active ingredients (artemisinin, paclitaxel, etc.) in medicinal plants [37], is regulated by AP2/ERF, which also controls plant growth and development and the primary metabolism. The flavonoid biosynthesis regulatory network, on the other hand, relies heavily on the bHLH family. In general, it is believed that NAC, MYB, and C2H2 families are involved in the control of primary and secondary metabolic biosynthesis in plants [38,39,40].

## 4. Discussion

According to classical herb book records, TH from SS is the most extensively utilized and has better efficacy [41,42,43], which is also consistent with our previous findings [16]. Meanwhile, other researchers have discovered that the synthesis and accumulation of the major active components including flavonoids and phenolic acids in TH from different hosts vary. The utilization of TH from different hosts, however, is severely constrained due to a lack of knowledge of the underlying molecular mechanism causing differences in the biosynthesis and accumulation of these components. In this work, we selected the most commonly used TH from SS and FXS and then carried out an integrated transcriptomic and metabolite profiling investigation.

A total of 81 compounds were identified in metabolite analysis, including 37 flavonoids, 7 organic acids, 5 tannins, 4 phenylpropanoids, and 18 others. As demonstrated in Figure 5, the overall chemical properties of TH from two hosts were similar, and the relative levels of metabolites in the two groups of samples had certain fluctuations. TH from SS had higher relative flavonoid aglycones and glycosides levels, as well as a higher proportion of flavan-3-ol and flavonol glycosides. Quercitrin, as a representative flavonoid in TH, had significant differences between the two groups. Based on the analysis of the overall change level of metabolites, this could provide basic data for the influence of host plants on the synthesis and accumulation of parasitic metabolites of TH, and also for the proteomics and transcription of parasitic metabolites of different host plants. The analysis of the overall change levels of metabolites could provide a basis for the influence of host plants on the synthesis and accumulation of metabolites of TH, and provide a scientific basis for the subsequent transcriptomic analysis. Meanwhile, since TH is a semi-parasitic plant, and apart from absorbing water and inorganic salts from the host plant, whether there is any mutual inhibition or promotion between it and the host plant in terms of its own growth and development is still an unknown topic that deserves further exploration [44]. 

Based on the fact that flavonoids are the main components exerting activity, we chose flavonoids for analysis in the transcriptomic study. Transcriptome studies are increasingly expanding from model plants to non-model plants, and plant gene sequence analysis has greatly expanded plant databases. It has been shown that differences in the expression of genes based on metabolic pathways can be used to investigate the regulation of secondary metabolite synthesis and accumulation at the molecular level during plant development [45]. At present, the whole chloroplast genome of TH has been sequenced, but because TH is not a model plant the study of transcriptome data of TH is limited. The annotated results revealed that 54.77% of the genes had no annotation information, indicating that the genetic information of TH needs to be mined and fully investigated, as well as substantially impeding the mining of important genes that regulate the synthesis and accumulation of active components. At the same time, since various types of bioinformatics databases are still being updated and enhanced, it is possible that certain specific new genes are yet to be found, necessitating more extensive research.

DEGs were not only involved in carbohydrate metabolism, such as in glycolysis/gluconeogenesis, pentose phosphate pathway, pentose and glucuronate interconversions, and starch and sucrose metabolism, but were also involved in lipid and amino acid metabolism such as tyrosine metabolism, cysteine methionine metabolism, pyrimidine metabolism, and phosphoinositol metabolism, as well as carbon fixation and other processes in photosynthesis. In addition, DEGs also participated in organic selenium compound metabolism, sulfur metabolism, keratin, amber and wax biosynthesis, and other processes. Some DEGs were involved in ubiquinone and other terpenoid-quinone biosynthesis, terpenoid backbone biosynthesis, brassinosteroid biosynthesis, and zeatin biosynthesis, and nine were related to flavonoid biosynthesis. Based on the above results, we speculated that in the early stage of parasitism, the host and parasite interact with each other through suckers, and TH changes to adapt to the host environment in order to survive better [46]. There were 79 genes annotated in the phenylpropanoid synthesis pathway, much higher than the number of genes annotated in the flavonoid synthesis pathway, which was the upstream of flavonoid synthesis, so further analysis of this pathway was needed in the later period. Studies have shown that Cuscuta has a lot of molecular communication with its host, and the host transfers a lot of mRNA to Cuscuta, which provides nutrition through degradation and might also be translated into proteins to function, and which ultimately affected the metabolic pathways in Cuscuta [47]. However, at present, there are few studies on the flow of proteins and mRNA between parasitism and the host. Therefore, whether there is similar transfer and communication of a large amount of mRNA and proteins between the host and TH is worth further exploration.

The identified genes were involved in the upstream synthesis of flavonoids, including *PAL*, *4CL*, and *CHI*. *PAL* employed phenylalanine, a prerequisite for the biosynthesis of phenylpropane, to produce cinnamic acid and coumaric acid. Coumaric acid is formed when *4CL* reacts with coumaroyl coenzyme A, which is then transferred to the flavonoid biosynthesis pathway via *CHS* [48]. *CHI* acts as a major enzyme in the flavonoid metabolic pathway that converts naringenin chalcone to naringenin, and acts as a bridge for the creation of other flavonoids. It plays a crucial role in biosynthesis involving various defense products, plant antibacterial mechanisms, and stress resistance [48]. The expression of *FLS* and *F3H*, as well as the aforementioned biosynthetic genes, did not differ significantly between the two groups. Additionally, the flavonoids’ related glycosides exhibited similar expression patterns. Multiple factors, including plant species, developmental stage, tissue site, and growth environment, work together to govern the synthesis and accumulation of flavonoid components in plants. Studies have shown that transcription factors such as R2R3-MYB and bHLH, as well as transcription factor families such as NAC, WRKY, and bZIP, regulate the expression of key genes at the transcriptional level, which in turn affects the synthesis and accumulation of flavonoids [49,50]. With notable exceptions only in the in the downstream of the phenylpropanoid biosynthesis pathway, the expression of genes ascribed to the upstream and downstream of the flavonoid biosynthesis pathway generally did not differ considerably from one another. The up- or down-regulation of the enzymes engaged in pathways might regulate downstream product synthesis either positively or negatively. This might be utilized to rationally explain that the TH from SS and FXS are undoubtedly not significantly different in terms of secondary metabolites. The possible reason for this phenomenon was that the enzyme system in the plant was in a less active state at the harvest time of the samples.

## 5. Conclusions

In summary, genes annotated to flavonoid biosynthesis in TH were identified by comparative transcriptomics. The pattern of gene expression in both groups was broadly consistent with the trend of accumulation of related compounds in the metabolome. However, the non-significant differences in gene expression between the two groups, combined with metabolomics, further illustrated that the differences between TH from two hosts may indeed be tiny. Our study combined metabolomics and transcriptomics to reveal the mechanism of the differences in the active constituents in TH from SS and FXS, which was helpful for the quality control and selection of hosts, and could also provide basic data for transcriptomic studies of TH from other hosts.

## Figures and Tables

**Figure 1 genes-14-01040-f001:**
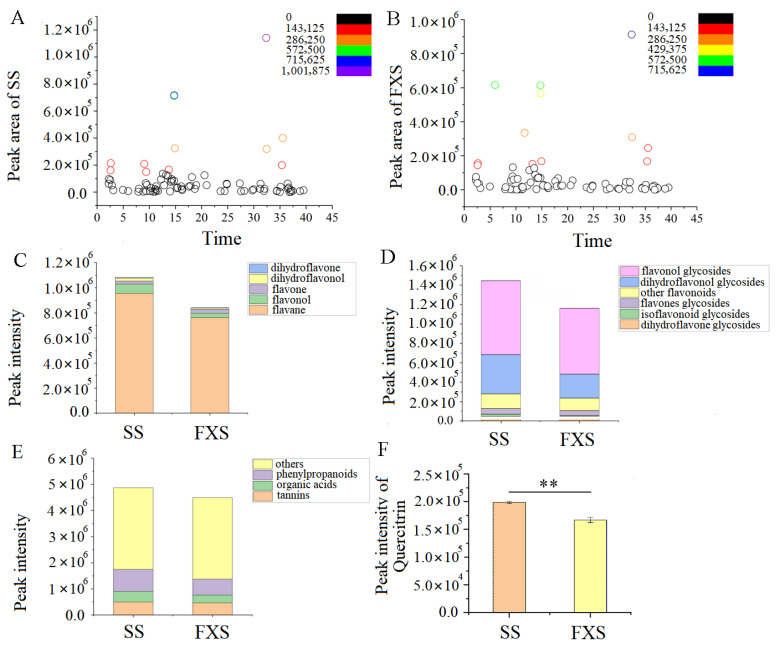
Results of metabolomics analysis. (**A**) Distribution of constituents identified in Taxilli Herba from SS; (**B**) distribution of constituents identified in Taxilli Herba from FXS; (**C**) total content changes in flavonoids; (**D**) total content changes in flavonoid glycosides; (**E**) total content changes in phenylpropanoids, organic acids, tannins, and others between two groups; and (**F**) the content change of quercitrin (“**’’: *p* < 0.01).

**Figure 2 genes-14-01040-f002:**
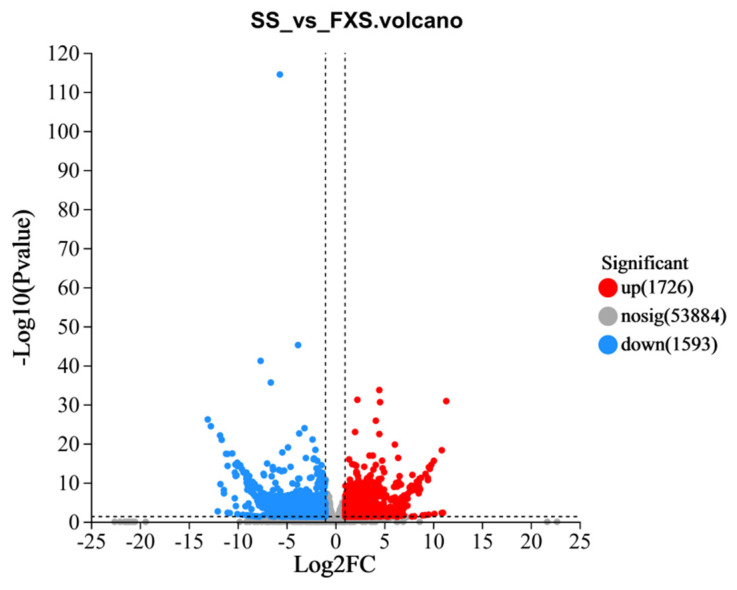
Volcano map analysis of DEGs (red points represent up-regulated genes, blue points represent down-regulated genes, and unchanged genes are gray).

**Figure 3 genes-14-01040-f003:**
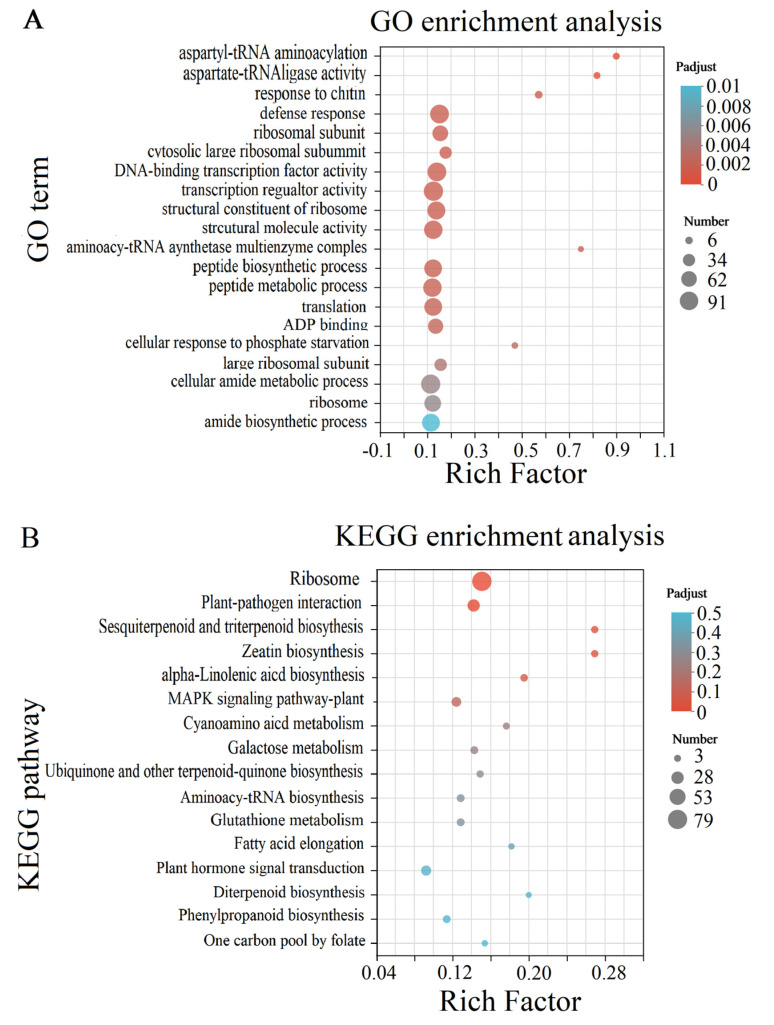
Functional annotation and enrichment analysis. (**A**) GO enrichment of top 20 terms between SS and FXS groups; (**B**) KEGG enrichment of top 20 pathways SS and FXS groups.

**Figure 4 genes-14-01040-f004:**
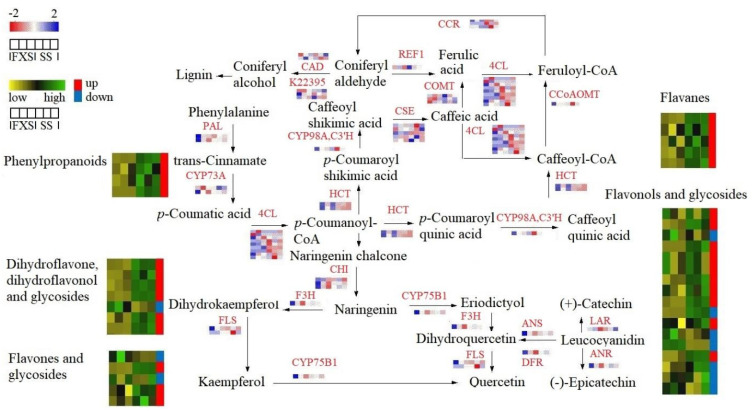
Putative network of biosynthetic pathways of flavonoids in Taxilli Herba.

**Figure 5 genes-14-01040-f005:**
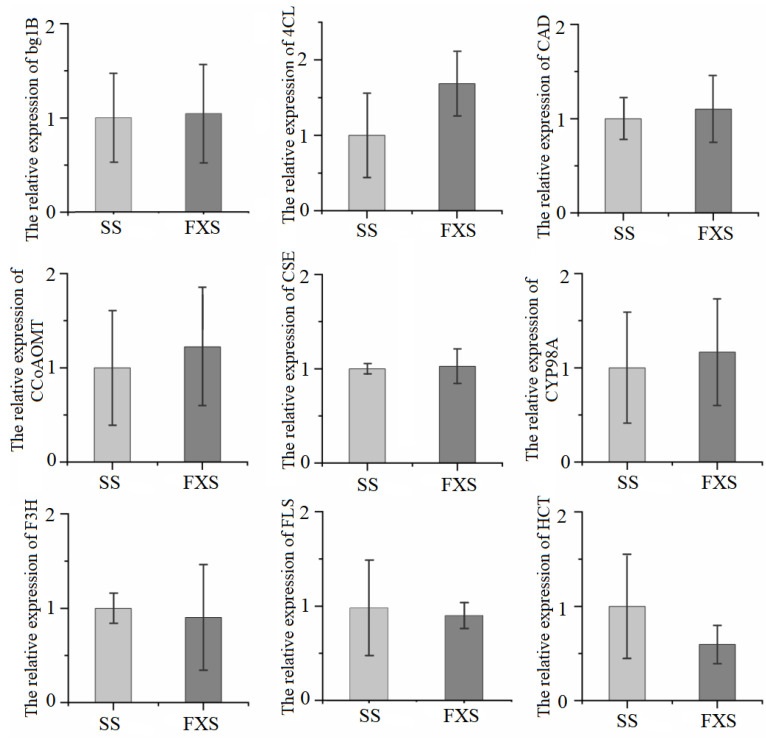
Expression patterns of genes (*bg1B*, *4CL*, *CAD*, *CCoAOMT*, *CSE*, *CYP98A*, *F3H*, *FLS*, *HCT*) between SS and FXS groups (The means ± SD).

## Data Availability

The data and materials supporting the conclusions of this study are included within the article and its additional files. All the raw reads generated in this study have been deposited in the NCBI with the BioProject accession number PRJNA904042.

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
