# Peer review of "Metabolite Profiling and Transcriptome Analysis Explain the Difference in Accumulation of Bioactive Constituents in Taxilli Herba from Two Hosts"

_genes, 2023, doi:10.3390/genes14051040_

Round 1

Reviewer 1 Report

In the manuscript "Metabolite Profiling and Transcriptome Analysis Explains Difference in Accumulation of Bioactive Constituents in Taxilli Herba from Two Hosts", the authors investigated the molecular mechanisms involved in the biosynthesis of the chemical components of TH from SS and FXS, using an integrated transcriptomic analysis and metabolomic approach. Bioinformatics analysis was used to select out and analyze differentially expressed genes (DEGs), some of the candidate genes were then verified through qPCR.

Overall the study is very interesting including comprehensive metabolomics and transcriptomics analyses. The introduction contains sufficient information regarding the background citing current literature. The methodology is completely described and results well-presented. 

The Discussion section should be extended a bit more. What the results demonstrate should be explained in detail. The implications of this study and research gaps identified should be mentioned.

Please correct the typo in the Title. Please give a thorough review to the whole text of manuscript and correct the typos.

Author Response

Dear Reviewer:

Thank you for your comments concerning the manuscript entitled “Metabolite Profiling and Transcriptome Analysis Explains Difference in Accumulation of Bioactive Constituents in Taxilli Herba from Two Hosts” (Manuscript ID: genes-2326185). Those comments are all valuable and very helpful for revising and improving our paper, as well as the important guiding significance to our researches. We have studied comments carefully and have made correction which we hope meet with approval. The “Track Changes” function in Microsoft word was used, the revised portion were clearly highlighted in the manuscript. The main corrections in the paper and the responses to the reviewers, comments are as following:

Response to Reviewer 1 Comments

In the manuscript "Metabolite Profiling and Transcriptome Analysis Explains Difference in Accumulation of Bioactive Constituents in Taxilli Herba from Two Hosts", the authors investigated the molecular mechanisms involved in the biosynthesis of the chemical components of TH from SS and FXS, using an integrated transcriptomic analysis and metabolomic approach. Bioinformatics analysis was used to select out and analyze differentially expressed genes (DEGs), some of the candidate genes were then verified through qPCR.

Overall the study is very interesting including comprehensive metabolomics and transcriptomics analyses. The introduction contains sufficient information regarding the background citing current literature. The methodology is completely described and results well-presented. 

Reply: First of all, thank you very much for your recognition and encouragement to our work. We appreciate for your warm work earnestly.

In order to fulfill the standards be set by Journal, we revised the article on the basis of reviewer’ comments. And hope that the correction will meet with approval.

Once again, thank you very much for your suggestions.

Point 1: The Discussion section should be extended a bit more. What the results demonstrate should be explained in detail. The implications of this study and research gaps identified should be mentioned.

Reply 1: Thanks for your professional comments. we have extended the Discussion section, explained the results demonstrate in detail, and mentioned the implications of this study and research gaps identified. The specific is as follows:

According to classical herb books’ records, TH from SS is the most extensively utilized and has better efficacy [41–43], which is also consistent with our previous findings [16]. Meanwhile, other researchers have discovered that the synthesis and accumulation of the major active components including flavonoids and phenolic acids in TH from different hosts vary. The utilization of TH from different hosts, however, is severely constrained due to a lack of knowledge of the underlying molecular mechanism causing differences in the biosynthesis and accumulation of these components. In this work, we selected the most commonly used TH from SS and FXS and then carried out an integrated transcriptomic and metabolite profiling investigation.

A total of 81 compounds were identified in metabolite analysis, including 37 flavonoids, 7 organic acids, 5 tannins, 4 phenylpropanoids, and 18 others. As demonstrated in Figure 5, the overall chemical properties of TH from two hosts were similar, and the relative levels of metabolites in the two groups of samples had certain fluctuations. TH from SS had higher relative flavonoid aglycones and glycosides levels, as well as a higher proportion of flavan-3-ol and flavonol glycosides. Quercitrin, as a representative flavonoid in TH, had significant differences between the two groups. Based on the analysis of the overall change level of metabolites, it could provide basic data for the influence of host plants on the synthesis and accumulation of parasitic metabolites of TH, and also for the proteomics and transcription of parasitic metabolites of different host plants. The analysis of the overall change level of metabolites could provide a basis for the influence of host plants on the synthesis and accumulation of metabolites of TH, and provide a scientific basis for subsequent transcriptomic analysis. Meanwhile, since TH is a semi-parasitic plant, apart from absorbing water and inorganic salts from the host plant, whether there is any mutual inhibition or promotion between it and the host plant in terms of its own growth and development is still an unknown topic that deserves further exploration [44].

Based on the fact that flavonoids are the main components exerting the activity, we chose flavonoids for analysis in the transcriptomic study. Transcriptome studies are increasingly expanding from model plants to non-model plants, and plant gene sequence analysis has greatly expanded plant databases. It has been shown that differences in the expression of genes based on metabolic pathways can be used to investigate the regulation of secondary metabolite synthesis and accumulation at the molecular level during plant development [45]. At present, the whole chloroplast genome of TH has been sequenced, but because TH is not a model plant, the study of transcriptome data of TH is limited. The annotated results revealed that 54.77% of the genes had no annotation information, indicating that the genetic information of TH needs to be mined and fully investigated, as well as substantially impeding the mining of important genes that regulate the synthesis and accumulation of active components. At the same time, since various types of bioinformatics databases are still being updated and enhanced, it is possible that certain specific new genes are still yet to be found, necessitating more extensive research.

DEGs were not only involved in carbohydrate metabolism such as glycolysis/gluconeogenesis, pentose phosphate pathway, pentose and glucuronate interconversions, starch and sucrose metabolism, but also involved in lipid and amino acid metabolism such as tyrosine metabolism, cysteine methionine metabolism, pyrimidine metabolism and phosphoinositol metabolism, as well as carbon fixation and other processes in photosynthesis. In addition, DEGs also participated in organic selenium compound metabolism, sulfur metabolism, keratin, amber and wax biosynthesis and other processes. Some of DEGs involved in ubiquinone and other terpenoid-quinone biosynthesis, terpenoid backbone biosynthesis, brassinosteroid biosynthesis, zeatin biosynthesis, and 9 were related to flavonoid biosynthesis. Based on the above results, we speculated that in the early stage of parasitism, the host and parasitism interact with each other through suckers, and TH changes to adapt to the host environment in order to survive better [46]. There were 79 genes annotated in the phenylpropanoid synthesis pathway, much higher than the number of genes annotated in the flavonoid synthesis pathway, which was the upstream of flavonoid synthesis, so further analysis of this pathway was needed in the later period. Studies have shown that Cuscuta has a lot of molecular communication with its host, and the host transfers a lot of mRNA to Cuscuta, which provides nutrition through degradation, and might also be translated into proteins to function, and ultimately affected the metabolic pathways in Cuscuta [47]. However, at present, there are few studies on the flow of proteins and mRNA between parasitism and the host. Therefore, it is worth further exploration whether there is similar transfer and communication of a large amount of mRNA and proteins between the host and TH.

The identified genes were involved in the upstream synthesis of flavonoids, including PAL, 4CL, and CHI. PAL employed phenylalanine, a prerequisite for the biosynthesis of phenylpropane, to produce cinnamic acid and coumaric acid. Coumaric acid is formed when 4CL reacts with coumaroyl coenzyme A, which is then transferred to the flavonoid biosynthesis pathway via CHS [48]. CHI acts as a major enzyme in the flavonoid metabolic pathway that converts naringenin chalcone to naringenin and acts as a bridge for the creation of other flavonoids. It plays a crucial role in biosynthesis involving various defense products, plant antibacterial mechanisms, stress resistance[48]. The expression of FLS and F3H, as well as the aforementioned biosynthetic genes, did not differ significantly between the two groups. Additionally, the flavonoids’ related glycosides exhibited similar expression patterns. Multiple factors, including plant species, developmental stage, tissue site, and growth environment, work together to govern the synthesis and accumulation of flavonoid components in plants. Studies have shown that transcription factors such as R2R3-MYB and bHLH as well as transcription factor families such as NAC, WRKY, and bZIP regulate the expression of key genes at the transcriptional level, which in turn affect the synthesis and accumulation of flavonoids [49,50]. With notable exceptions only in the in the downstream of the phenylpropanoid biosynthesis pathway, the expression of genes ascribed to the upstream and downstream of the flavonoid biosynthesis pathway generally did not differ considerably from one another. Up- or down-regulation of enzymes engaged in pathways might regulate downstream product synthesis either positively or negatively. This might be utilized to rationally explain that the TH from SS and FXS are undoubtedly not significantly different in terms of secondary metabolites. The possible reason for this phenomenon was that the enzyme system in the plant is in a less active state at the harvest time of samples.

Point 2: Please correct the typo in the Title. Please give a thorough review to the whole text of manuscript and correct the typos.

Reply 2: Thanks for your professional advice. We have corrected the typo in the Title and check the full manuscript and corrected the typos.

We have checked and corrected other errors in the full text.

Once again, thank you very much for your comments and suggestions.

We appreciate for Editors/Reviewers’ warm work earnestly, and hope that the correction will meet with approval. Thank you very much.

Yours sincerely,

Jiahuan Yuan

Reviewer 2 Report

The English form of the MS should be highly revised.. the title include a work with no sense "TOST". What is this? Even in the title the verb should be plural and not in 3rd person. In line 12 "Flavonoids IS..." it should sound "Flavonoids ARE..". This makes very difficult the real comprehension of the paper. An intense revision of the language is needed.

Fig 6, in general, no statically significant differences can be detected in the studied genes. How can the authors explain this evidence? Why did they select exactly them.. they should specify this aspect, as one would expect that the authors select those genes that are considered to be potentilly modulated to confirm their Omics analysis.

A paragraph on statistics should be reported in M&M section. Line 216, how was calculated the p value reported here? Fig 5, were the results reported here evaluated in statical terms?

The term quercetin was written in different way in the text, please homogenise it!

The discussion is very limited as paragraph. I would suggest to incorporate it with results. Morevoer, the data obtained in this paper should be commented better with respect to the literature. I know that the specific case the authors are investigating has been never documented in the literature but they could find similar aspect with other case studies, in order to support or criticize their own results. For instance, they could put in parallel the effect on the synthesis of secondary metabolites when exposed to a pathogen and the concept of the allelopathy that could represent the key to explain the communication mechanisms between the studied plant and the host. I mention here some papers that should be all mentioned to reinforce their text, as reported before: Microbial pathogenesis 124 (2018): 198-202; Journal of plant research 132 (2019): 439-455; Frontiers in Plant Science 7 (2016): 594; Allelopathy: a physiological process with ecological implications. Springer Science & Business Media, 2006.

Author Response

Dear Reviewer:

Thank you for your comments concerning the manuscript entitled “Metabolite Profiling and Transcriptome Analysis Explains Difference in Accumulation of Bioactive Constituents in Taxilli Herba from Two Hosts” (Manuscript ID: genes-2326185). Those comments are all valuable and very helpful for revising and improving our paper, as well as the important guiding significance to our researches. We have studied comments carefully and have made correction which we hope meet with approval. The “Track Changes” function in Microsoft word was used, the revised portion were clearly highlighted in the manuscript. The main corrections in the paper and the responses to the reviewers, comments are as following:

Response to Reviewer 2 Comments

Point 1: The English form of the MS should be highly revised. the title include a work with no sense "TOST". What is this? Even in the title the verb should be plural and not in 3rd person. In line 12 "Flavonoids IS..." it should sound "Flavonoids ARE..". This makes very difficult the real comprehension of the paper. An intense revision of the language is needed.

Reply 1: Thanks for your professional advice. We have tried our best to improve the English form of the full text at the aid of a native English-speaker to make it easier to understand the paper. We hope that the correction would meet with approval.

(1) The title including a word with no sense “TOST” was caused by our spelling mistakes and we have revised “TOST” in the title into “Hosts”.

(2) We have revised “Explains” into “Explain”.

(3) In line 12 “Flavonoids IS...”, we have revised “IS” into “are”.

Point 2: Fig 6, in general, no statically significant differences can be detected in the studied genes. How can the authors explain this evidence? Why did they select exactly them.. they should specify this aspect, as one would expect that the authors select those genes that are considered to be potentilly modulated to confirm their Omics analysis.

Reply 2: Thanks for your professional comments. 

  1. We revalidated the original data of the relative expressions (as shown in the table below) and the results showed that there is no statically significant differences can be detected in the studied genes [1].

Table The original data of genes expression 

Gene ID

Groups

ACTIN CÑ‚

Target Gene CÑ‚

TRINITY_DN5837_c0_g1  (CYP98A)

SS

SS1

26.80063629

22.85618591

SS2

26.85132027

21.58279419

SS3

26.6302948

22.78386879

FXS

FXS1

27.21929741

22.43600464

FXS2

27.76563263

22.66190147

FXS3

27.72080803

23.87590218

TRINITY_DN9022_c0_g1 (bglB)

SS

SS1

26.80063629

25.46796227

SS2

26.85132027

26.8691082

SS3

26.6302948

25.77772331

FXS

FXS1

27.21929741

26.57385445

FXS2

27.76563263

27.65450287

FXS3

27.72080803

26.16615868

TRINITY_DN3782_c0_g1 (CAD)

SS

SS1

26.80063629

23.52405357

SS2

26.85132027

22.91182518

SS3

26.6302948

23.28820801

FXS

FXS1

27.21929741

23.43699074

FXS2

27.76563263

24.17874718

FXS3

27.72080803

24.17790985

TRINITY_DN192_c0_g1 (CSE)

SS

SS1

26.80063629

24.01654816

SS2

26.85132027

24.0076046

SS3

26.6302948

23.87521362

FXS

FXS1

27.21929741

24.4904232

FXS2

27.76563263

24.75722504

FXS3

27.72080803

25.00432396

TRINITY_DN6080_c0_g1 (CCoAOMT)

SS

SS1

26.59055138

22.77814484

SS2

26.0169735

21.38600349

SS3

26.10632896

22.52182579

FXS

FXS1

26.77087402

22.44122887

FXS2

27.7872467

22.71881104

FXS3

27.46388245

23.97188759

TRINITY_DN3882_c0_g1 (HCT)

SS

SS1

26.59055138

26.91615868

SS2

26.0169735

27.68909264

SS3

26.10632896

28.39130592

FXS

FXS1

26.77087402

28.99976349

FXS2

27.7872467

29.38236046

FXS3

27.46388245

29.95077324

TRINITY_DN1730_c0_g1 (FLS)

SS

SS1

26.59055138

21.01255417

SS2

26.0169735

22.32076454

SS3

26.10632896

20.79281807

FXS

FXS1

26.77087402

24.31616783

FXS2

27.7872467

22.39328766

FXS3

27.46388245

25.71079445

TRINITY_DN29174_c0_g1 (4CL)

SS

SS1

26.29935646

24.34881783

SS2

25.55464745

25.48370361

SS3

27.62908745

23.85072327

FXS

FXS1

26.22473145

24.7548275

FXS2

26.50338936

24.31330299

FXS3

28.83081055

24.07038879

TRINITY_DN3827_c0_g1 (F3H)

SS

SS1

26.59055138

25.2186203

SS2

26.0169735

25.63663673

SS3

26.10632896

25.55818939

FXS

FXS1

26.77087402

26.61763

FXS2

27.7872467

26.86494255

FXS3

28.83081055

28.86357117

Tang, H.; Zhang, M.; Liu, J.Y.; Cai, J. Metabolomic and transcriptomic analyses reveal the characteristics of tea flavonoids and caffeine accumulation and regulation between Chinese carieties (Camellia sinensis var. sinensis) and assam varieties (C. sinensis var. assamica). Genes. 2022, 13, 1994.

  1. We selected these genes for the following reasons:

(1) Flavonoids are the main active constituents in TH, and these genes were annotated to the pathway of flavonoid biosynthesis with certain biological significance in our research.

(2) The FPKM of these genes was greater than 50 in at least one sample.

(3) The read count of these genes is relatively high.  

(4) The fold changes of the relative expression of these genes were relatively high with lower p-value.

Point 3: A paragraph on statistics should be reported in M&M section. Line 216, how was calculated the p value reported here? Fig 5, were the results reported here evaluated in statical terms?

Reply 3: Thanks for your professional advice.

(1) We have added a paragraph on statistics in M&M section, and the details are as follows (Line 187-191 of Page 5 in revised version):

2.3.6. Data Processing

The graphs were created by Origin (version 2022). The experimental data were analyzed by t-test (SPSS, version 21.0) and the p-value < 0.05 indicated significant difference. The results were represented by the mean values and standard deviations of three biological replicates.

(2) Line 216 (Line 225 in revised version), the p-value was calculated by importing the data into SPSS (version 21.0) and then conducting t-test.

(3) Fig 5, the results reported here were evaluated in statical terms.

The description of the data that made up the flavonoids heatmap was described in line 201-203: “The relative peak area of the identified metabolites was extracted and the average of the relative peak area of three independent samples in each group was calculated as the relative content of a metabolite. ”

  The description of the data of the expression level of genes was described in line line 166-168: “The expression level of each transcript was calculated according to the fragments per kilobases per million reads (FPKM) and the abundance was quantified using RSEM software.

We have made the appropriate modifications to Figure 5, as shown below:

Figure 5. Putative network of biosynthetic pathway of flavonoids in Taxilli Herba.

Point 4: The term quercetin was written in different way in the text, please homogenise it!

Reply 4: Thanks for your kind reminder. We have checked the full text and homogenise the term quercetin.

Point 5: The discussion is very limited as paragraph. I would suggest to incorporate it with results. Moreover, the data obtained in this paper should be commented better with respect to the literature. I know that the specific case the authors are investigating has been never documented in the literature but they could find similar aspect with other case studies, in order to support or criticize their own results. For instance, they could put in parallel the effect on the synthesis of secondary metabolites when exposed to a pathogen and the concept of the allelopathy that could represent the key to explain the communication mechanisms between the studied plant and the host. I mention here some papers that should be all mentioned to reinforce their text, as reported before: Microbial pathogenesis 124 (2018): 198-202; Journal of plant research 132 (2019): 439-455; Frontiers in Plant Science 7 (2016): 594; Allelopathy: a physiological process with ecological implications. Springer Science & Business Media, 2006.

Reply 5:  Thanks for your professional advice.

(1) We attached great importance to your comments and tried to incorporate the discussion part with the results, but this made it difficult to understand what we want to express finally. Therefore, we separated the result from the discussion part. At the same time, we have added relevant literature to support our results.  

(2) The purpose of our study was to investigate the differences in metabolism and transcription levels of TH from two hosts and investigate the relationship between the regulation of genes expression and the accumulation of bioactive constituents rather than the communication mechanisms between hosts and TH. Meanwhile, we have carefully read the literature you mentioned and made appropriate references in the Discussion Section.

We have revised the discussion part in detail according to your comments. The details are as follows:

According to classical herb books’ records, TH from SS is the most extensively utilized and has better efficacy [41–43], which is also consistent with our previous findings [16]. Meanwhile, other researchers have discovered that the synthesis and accumulation of the major active components including flavonoids and phenolic acids in TH from different hosts vary. The utilization of TH from different hosts, however, is severely constrained due to a lack of knowledge of the underlying molecular mechanism causing differences in the biosynthesis and accumulation of these components. In this work, we selected the most commonly used TH from SS and FXS and then carried out an integrated transcriptomic and metabolite profiling investigation.

A total of 81 compounds were identified in metabolite analysis, including 37 flavonoids, 7 organic acids, 5 tannins, 4 phenylpropanoids, and 18 others. As demonstrated in Figure 5, the overall chemical properties of TH from two hosts were similar, and the relative levels of metabolites in the two groups of samples had certain fluctuations. TH from SS had higher relative flavonoid aglycones and glycosides levels, as well as a higher proportion of flavan-3-ol and flavonol glycosides. Quercitrin, as a representative flavonoid in TH, had significant differences between the two groups. Based on the analysis of the overall change level of metabolites, it could provide basic data for the influence of host plants on the synthesis and accumulation of parasitic metabolites of TH, and also for the proteomics and transcription of parasitic metabolites of different host plants. The analysis of the overall change level of metabolites could provide a basis for the influence of host plants on the synthesis and accumulation of metabolites of TH, and provide a scientific basis for subsequent transcriptomic analysis. Meanwhile, since TH is a semi-parasitic plant, apart from absorbing water and inorganic salts from the host plant, whether there is any mutual inhibition or promotion between it and the host plant in terms of its own growth and development is still an unknown topic that deserves further exploration [44].

Based on the fact that flavonoids are the main components exerting the activity, we chose flavonoids for analysis in the transcriptomic study. Transcriptome studies are increasingly expanding from model plants to non-model plants, and plant gene sequence analysis has greatly expanded plant databases. It has been shown that differences in the expression of genes based on metabolic pathways can be used to investigate the regulation of secondary metabolite synthesis and accumulation at the molecular level during plant development [45]. At present, the whole chloroplast genome of TH has been sequenced, but because TH is not a model plant, the study of transcriptome data of TH is limited. The annotated results revealed that 54.77% of the genes had no annotation information, indicating that the genetic information of TH needs to be mined and fully investigated, as well as substantially impeding the mining of important genes that regulate the synthesis and accumulation of active components. At the same time, since various types of bioinformatics databases are still being updated and enhanced, it is possible that certain specific new genes are still yet to be found, necessitating more extensive research.

DEGs were not only involved in carbohydrate metabolism such as glycolysis/gluconeogenesis, pentose phosphate pathway, pentose and glucuronate interconversions, starch and sucrose metabolism, but also involved in lipid and amino acid metabolism such as tyrosine metabolism, cysteine methionine metabolism, pyrimidine metabolism and phosphoinositol metabolism, as well as carbon fixation and other processes in photosynthesis. In addition, DEGs also participated in organic selenium compound metabolism, sulfur metabolism, keratin, amber and wax biosynthesis and other processes. Some of DEGs involved in ubiquinone and other terpenoid-quinone biosynthesis, terpenoid backbone biosynthesis, brassinosteroid biosynthesis, zeatin biosynthesis, and 9 were related to flavonoid biosynthesis. Based on the above results, we speculated that in the early stage of parasitism, the host and parasitism interact with each other through suckers, and TH changes to adapt to the host environment in order to survive better [46]. There were 79 genes annotated in the phenylpropanoid synthesis pathway, much higher than the number of genes annotated in the flavonoid synthesis pathway, which was the upstream of flavonoid synthesis, so further analysis of this pathway was needed in the later period. Studies have shown that Cuscuta has a lot of molecular communication with its host, and the host transfers a lot of mRNA to Cuscuta, which provides nutrition through degradation, and might also be translated into proteins to function, and ultimately affected the metabolic pathways in Cuscuta [47]. However, at present, there are few studies on the flow of proteins and mRNA between parasitism and the host. Therefore, it is worth further exploration whether there is similar transfer and communication of a large amount of mRNA and proteins between the host and TH.

The identified genes were involved in the upstream synthesis of flavonoids, including PAL, 4CL, and CHI. PAL employed phenylalanine, a prerequisite for the biosynthesis of phenylpropane, to produce cinnamic acid and coumaric acid. Coumaric acid is formed when 4CL reacts with coumaroyl coenzyme A, which is then transferred to the flavonoid biosynthesis pathway via CHS [48]. CHI acts as a major enzyme in the flavonoid metabolic pathway that converts naringenin chalcone to naringenin and acts as a bridge for the creation of other flavonoids. It plays a crucial role in biosynthesis involving various defense products, plant antibacterial mechanisms, stress resistance[48]. The expression of FLS and F3H, as well as the aforementioned biosynthetic genes, did not differ significantly between the two groups. Additionally, the flavonoids’ related glycosides exhibited similar expression patterns. Multiple factors, including plant species, developmental stage, tissue site, and growth environment, work together to govern the synthesis and accumulation of flavonoid components in plants. Studies have shown that transcription factors such as R2R3-MYB and bHLH as well as transcription factor families such as NAC, WRKY, and bZIP regulate the expression of key genes at the transcriptional level, which in turn affect the synthesis and accumulation of flavonoids [49,50]. With notable exceptions only in the in the downstream of the phenylpropanoid biosynthesis pathway, the expression of genes ascribed to the upstream and downstream of the flavonoid biosynthesis pathway generally did not differ considerably from one another. Up- or down-regulation of enzymes engaged in pathways might regulate downstream product synthesis either positively or negatively. This might be utilized to rationally explain that the TH from SS and FXS are undoubtedly not significantly different in terms of secondary metabolites. The possible reason for this phenomenon was that the enzyme system in the plant is in a less active state at the harvest time of samples.

We have checked and corrected other errors in the full text.

Once again, thank you very much for your comments and suggestions.

We appreciate for Editors/Reviewers’ warm work earnestly, and hope that the correction will meet with approval. Thank you very much.

Yours sincerely,

Jiahuan Yuan
